# Socioeconomic Inequalities in Chronic Liver Diseases and Cirrhosis Mortality in European Urban Areas before and after the Onset of the 2008 Economic Recession

**DOI:** 10.3390/ijerph18168801

**Published:** 2021-08-20

**Authors:** Carme Borrell, Laia Palència, Lucia Bosakova, Mercè Gotsens, Joana Morrison, Claudia Costa, Dagmar Dzurova, Patrick Deboosere, Michala Lustigova, Marc Marí-Dell’Olmo, Sophia Rodopoulou, Paula Santana

**Affiliations:** 1Agència de Salut Pública de Barcelona, 08023 Barcelona, Spain; 2CIBER Epidemiología y Salud Pública (CIBERESP), 20029 Madrid, Spain; lpalenci@aspb.cat (L.P.); mmari@aspb.cat (M.M.-D.); 3Institut d’Investigació Biomèdica (IIB Sant Pau), 08003 Barcelona, Spain; mgotsens@aspb.cat; 4Department of Experimental Sciences and Health, Universitat Pompeu Fabra, 08003 Barcelona, Spain; 5Service of Health Information Systems, Agència de Salut Pública de Barcelona, 08023 Barcelona, Spain; 6Department of Health Psychology and Research Methodology, Medical Faculty, P. J. Safarik University of Kosice, 04180 Kosice, Slovakia; lucia.bosak@gmail.com; 7Olomouc University Social Health Institute (OUSHI), Palacky University in Olomouc, CZ-77900 Olomouc, Czech Republic; 8Service of Prevention and Care of Drug Addictions, Agència de Salut Pública de Barcelona, 08023 Barcelona, Spain; 9Institute of Health Equity at the Research Department of Epidemiology and Public Health, University College London, London WC1E 6BT, UK; j.morrison@ucl.ac.uk; 10Centre of Studies in Geography and Spatial Planning (CEGOT), University of Coimbra, 3000-370 Coimbra, Portugal; claudiampcosta@gmail.com (C.C.); paulasantana.coimbra@gmail.com (P.S.); 11Faculty of Science, Charles University, 116 36 Prague, Czech Republic; dagmar.dzurova@natur.cuni.cz (D.D.); michala.lustigova@gmail.com (M.L.); 12Department of Sociology, Interface Demography, Vrije Universiteit Brussel, 1050 Brussels, Belgium; Patrick.Deboosere@vub.ac.be; 13Service of Environmental Quality and Interventions, Agència de Salut Pública de Barcelona, 08023 Barcelona, Spain; 14Department of Hygiene, Epidemiology and Medical Statistics, Medical School, National and Kapodistrian University of Athens, 106 79 Athens, Greece; srodopoyl@med.uoa.gr; 15Department of Geography and Tourism, University of Coimbra, 3000-370 Coimbra, Portugal

**Keywords:** chronic liver diseases, liver cirrhosis, mortality, inequalities, urban areas, financial crisis

## Abstract

Objective: To analyse the trends in chronic liver diseases and cirrhosis mortality, and the associated socioeconomic inequalities, in nine European cities and urban areas before and after the onset of the 2008 financial crisis. Methods: This is an ecological study of trends in three periods of time: two before (2000–2003 and 2004–2008), and one after (2009–2014) the onset of the economic crisis. The units of analysis were the geographical areas of nine cities or urban areas in Europe. We analysed chronic liver diseases and cirrhosis standardised mortality ratios, smoothing them with a hierarchical Bayesian model by each city, area, and sex. An ecological regression model was fitted to analyse the trends in socioeconomic inequalities, and included the socioeconomic deprivation index, the period, and their interaction. Results: In general, chronic liver diseases and cirrhosis mortality rates were higher in men than in women. These rates decreased in all cities during the financial crisis, except among men in Athens (rates increased from 8.50 per 100,000 inhabitants during the second period to 9.42 during the third). Socioeconomic inequalities in chronic liver diseases and cirrhosis mortality were found in six cities/metropolitan areas among men, and in four among women. Finally, in the periods studied, such inequalities did not significantly change. However, among men they increased in Turin and Barcelona and among women, several cities had lower inequalities in the third period. Conclusions: There are geographical socioeconomic inequalities in chronic liver diseases and cirrhosis mortality, mainly among men, that did not change during the 2008 financial crisis. These results should be monitored in the long term.

## 1. Introduction

Different studies showed that during the economic recession that started in 2008, some health indicators deteriorate and others remained stable [1,2,3]. Analyses of the effects of financial crises on inequalities in health and mortality led to heterogeneous results, with inequalities increasing in some countries and remaining stable in others [1,4,5].

Cirrhosis and other chronic liver diseases are a major and preventable cause of death worldwide; the Global Burden of Disease Study 2017 described these diseases as the 13th leading cause of years of life lost globally [6,7]. Variations in cirrhosis mortality between countries reflect differences in risk factors, such as alcohol use and hepatitis B and C infection [8]. Several studies have described inequalities in mortality due to cirrhosis [9,10,11,12] and alcohol use [13,14,15]. However, few studies have analysed trends in liver-related and cirrhosis mortality after the start of the economic recession of 2008 [7,16,17,18]. Risk factors of cirrhosis mortality may change during economic recessions and influence the trends and socioeconomic inequalities in chronic liver diseases and cirrhosis mortality. For alcohol consumption, there is a contradictory evidence on the impact of economic crises on this consumption and alcohol-related-health problems. Possible mechanisms related to how these crisis affect alcohol consumption have been described as psychological distress, triggered by unemployment and income reductions, that increases alcohol consumption and secondly tighter budget constraints than imply less money spent in alcoholic beverages and reduce consumption [19]. Hepatitis B and C affect vulnerable populations such as for example persons who inject drugs or migrants of low income countries, populations who have suffered the financial crisis [20].

Therefore, the objective of this study was to analyse the trends in chronic liver diseases and cirrhosis mortality, and in socioeconomic inequalities in this mortality, in nine European urban areas before and after the onset of the 2008 financial crisis.

## 2. Methods

### 2.1. Design, Units of Analysis, and Study Population

This is an ecological study of trends in three periods of time: two periods before the 2008 economic crisis and one during the crisis (2000–2003, 2004–2008 and 2009–2014). This study is part of the EURO-HEALTHY project (funded by Horizon 2020: http://www.euro-healthy.eu), through which other causes of death have been described [21,22]. The units of analysis were the geographical areas of nine European cities or metropolitan areas (Athens metropolitan area, Barcelona city, Berlin-Brandenburg metropolitan region, Brussels metropolitan area, Lisbon metropolitan area, Greater London, Prague city, Stockholm metropolitan area, and Turin city). The study population consisted of the individual residents in these areas during the three periods mentioned above.

### 2.2. Data and Measures

The data and measures used were the following:Mortality, corresponding to the number of deaths due to chronic liver diseases and cirrhosis (International Classification of Diseases (ICD)–9: 571; ICD–10: K70 and K74) by area, age group, and sex. Data were obtained from registries provided by local and national statistics institutions. The indicator of chronic liver diseases and cirrhosis mortality used for the analysis was the standardised mortality ratio (SMR), calculated by dividing the observed number of deaths in the study areas by the expected number of deaths (indirect method of age standardization). The latter was calculated considering the age-specific mortality rates in the standard population of the European Union (EU)-28 in the year 2007, and the population of the city/area by age groups. For descriptive purposes, we calculated the crude mortality rate (MR), and the indirectly standardised mortality rate (ISMR). The latter was calculated by multiplying the SMR in the city/metropolitan area by the crude rate in the standard population.Population, corresponding to the number of inhabitants living in the study areas. Most urban areas had data for the entire period studied or for at least 2 years. These data were stratified by age (5-year groups) and sex and were obtained from census or population registries. The median population of the areas ranged from 271 (Turin) to 193,630 (Berlin) (Table 1).Socioeconomic data, obtained from the census, consisting of several indicators selected to identify the level of deprivation of the areas studied. We built a composite socioeconomic deprivation index using principal component analysis within each city. The variables included in the analysis for 2001 (2002, in the case of Berlin) were the percentages of: unemployed population (≥16 years, economically active population); manual workers (≥16 years); population with primary education as the highest attainment (International Standard Classification of Education 0 and 1, except for London, which was 0, 1, and 2) (25–64 years); and population with university education (25–64 years). The percentage of manual workers was not available for Stockholm and the index was calculated with the other three indicators. The index of deprivation was the first component of the principal component analysis, which was performed separately for each city; therefore, it was possible to show the pattern of deprivation in each setting. The proportion of variance explained by the first component ranged from 0.578 in Lisbon to 0.918 in the Athens metropolitan area. The principal component analysis was performed using the prcomp function of the R statistical software package. The calculation was performed by a singular value decomposition of the data matrix, which is generally the preferred method for numerical accuracy.

### 2.3. Data Analysis

Due to the excessively variable estimates for the small areas included in this study, we smoothed the SMR and we used the hierarchical Bayesian model proposed by Besag, York and Mollié (BYM) [23]. This model takes two types of random effects into account: spatial and heterogeneous; the former takes into account of the spatial structure of the data, while the latter deals with non-structural (non-spatial) variability. We estimated smoothed SMR (sSMR) for both sexes and periods using the following model:(1)$$\left.\begin{array}{c} O_{i}\;\sim \;Poisson\left(E_{i}\theta_{i}\right)\\ \log\left(\theta_{i}\right)\;\sim \;\alpha +S_{i}+H_{i}\; \end{array}\right\}\;\left(model\;1\right)$$
where, for each area *i*, *O_i_* is the number of observed cases, *E_i_* the number of expected cases, *θ_i_*, the sSMR with respect to the European population, *S_i_* the spatial effect, and *H_i_* the heterogeneous effect. Let *α* be the intercept of the model. Expected cases were calculated by indirect standardisation, taking the liver cirrhosis mortality rates of the EU-28 in 2007 as reference (using the year approximately in the middle of the period) by age (using 5-year groups).

Moreover, to analyse the trend in socioeconomic inequalities, we fitted an ecological regression model, which included the socioeconomic deprivation index (through the continuous variable *D*), the period (through two dummy variables, P_2_ and P_3_), and their interaction:(2)$$\left.\begin{array}{c} O_{it}\;\sim \;Poisson\left(E_{it}\theta_{it}\right)\\ \log\left(\theta_{it}\right)\;\sim \;\alpha +\beta_{1}D_{i}+\beta_{2}P_{2t}+\beta_{3}P_{3t}+\beta_{4}P_{2t}D_{i}+\beta_{5}P_{3t}D_{i}+S_{it}+H_{it}\; \end{array}\right\}\;\left(model\;2\right)$$
where, for each area *i* and period *t* (*t* = 1 for the first period, *t* = 2 for the second period and *t* = 3 for the third period), *O_it_* is the number of observed cases, *E_it_* the number of expected cases, *θ_it_* the sSMR with respect to the European population, *D*_i_ the value of the deprivation index; *S_it_* the spatial effect, and *H_it_* the heterogeneous effect. Finally, *P_2t_* and *P_3t_* took the following values: *P_jt_* = 1 if *j = t*, and *P_jt_ = 0* if *j ≠ t*. Let *α* be the intercept of the model and *β_1_, β_2,_ β_3,_ β_4_,* and *β_5_* the parameters or coefficients associated with the different variables and their interactions. The expected cases were calculated as in the previous model. Changes between periods in the relationship between socioeconomic deprivation index and mortality were evaluated through the interactions included in model 2. Specifically, we studied the change between the first and second period (*β_4_*) and the second and third period (*β_5_–β_4_*).

In the two models (models 1 and 2), an intrinsic conditional autoregressive prior distribution was assigned to the spatial effect, which assumed that the expected value of each area coincided with the mean of the spatial effect of the adjacent areas and had a variance of σ_s_^2^, while the heterogeneous effect was represented using independent normal distributions with mean 0 and variance σ_h_^2^. A uniform distribution U (0,∞) was assigned to the standard deviations σ_s_ and σ_h_. A normal vague prior distribution was assigned to the parameters *α, β_1_, β_2_, β_3_, β_4,_ and β_5_.*

Given that the socioeconomic deprivation index scale is dimensionless and arbitrarily fixed, we calculated the relative risk (RR) for each city/metropolitan area, which compares liver cirrhosis mortality of the 95^th^ percentile value of socioeconomic deprivation (severe deprivation) to its 5^th^ percentile value (low deprivation). RR estimates were obtained based on the mean of their subsequent distribution, along with the corresponding 95% credible intervals (95% CI).

All analyses were performed using the Integrated Nested Laplace Approximation library of the R statistical package.

## 3. Results

Table 1 shows the characteristics of the nine areas studied. Greater London (7,322,403 inhabitants) is the area with the largest population size, and Turin (891,769 inhabitants) is the smallest. In the majority of areas, the median population (p50) by sex was under 6000, except districts in Berlin (median population of approximately 100,000) and municipalities in Athens (median population of approximately 30,000). Of note, the first period included 4 years (2000–2003), and the second and third period, 5 years (2004–2008 and 2009–2013, respectively) for most cities.

The description of mortality data is presented in Table 2. Chronic liver diseases and cirrhosis mortality rates (crude and age-adjusted) were higher among men than among women in the three periods across all cities. Rates were lower in Athens, Stockholm, and London (and in Lisbon for women only), and higher in Berlin. During the third period, death rates decreased in all settings except in Athens among men, where the ISMR increased from 8.50 per 100,000 inhabitants during the second period to 9.42 during the third.

Figure 1 and Figure 2 show the RR of the association between socioeconomic deprivation index and mortality due to chronic liver diseases and cirrhosis, for men and women during the three periods in each city or metropolitan area. Among men, RR was always higher than 1 in all cities, except for Prague, Brussels, and Berlin. The highest RR in the first period was found in Stockholm (RR = 6.24); however, it decreased over time (in the last period, RR = 3.86). Turin’s RR showed an increasing trend, with the highest value observed in the last period (RR = 4.06). Barcelona had a very similar increasing trend to that of Turin. However, these differences in RR did not have statistical significance.

Among women, a positive significant association between deprivation and mortality was identified in Turin, Stockholm, London, and Barcelona. During the third period, the RR only increased in London, although without statistical significance (RR in the second period: 1.74, 95% CI: 1.29–2.29; RR in the third period: 2.09, 95% CI: 1.59–2.71). In the other cities, instead, the RR decreased during this period (Athens, Barcelona, Lisbon, Stockholm, and Turin) although without statistical significance. The decrease in RR in Lisbon was noteworthy (RR in the second period: 1.80, 95% CI: 0.98–3.09; RR in the third period: 0.97, 95% CI: 0.49–1.76).

## 4. Discussion

This study found that mortality rates of chronic liver diseases and cirrhosis mortality decreased in all cities during the financial crisis period, except among men in Athens. Mortality rates in men were higher than those in women. Moreover, socioeconomic inequalities in liver cirrhosis mortality were found in 6 cities/metropolitan areas among men, and in 4 among women. Although changes in inequalities were not statistically significant among men, these increased in Turin and Barcelona. Among women, several cities had lower RR in the third period.

Previous studies on the long-term trends in cirrhosis mortality showed it decreased in the last four decades in the majority of countries in Western Europe, except for the UK, Ireland, and Finland [8,24,25,26,27]. This decline was mainly due to a decrease in alcohol consumption and an improvement in alcohol quality. Indeed, in EU member states, around 60–80% of deaths from liver disease are due to excessive alcohol consumption [2,3,4]. However, some countries in Eastern Europe had higher rates of cirrhosis-related mortality. Such mortality mainly increased during the 1980s and early 1990s as a result of the dissolution of the Union of Soviet Socialist Republics and the use of low-quality alcohol; more recently, rates decreased in the majority of these countries [25,26]. In Central Europe, cirrhosis-related mortality rates were high with a stable trend [8]. In our analysis, the only city where the rates did not decrease was Athens, where the financial crisis was particularly impactful. On this note, an increase in cirrhosis mortality and alcohol consumption in the Greek population aged 15–49 has been presented by the Global Burden of Diseases project (Greece) [28]. Bosque-Prous et al. carried out a study in 2006–2013 on alcohol consumption among adults aged 50–64 and found the prevalence of hazardous drinking decreased and abstention from alcohol consumption increased in the majority of European countries, except for the Czech Republic, where the opposite trends were observed among men [29]. However, the review carried out by Dom et al. [30] found mixed results, mainly related to a general decrease in alcohol consumption and an increase in harmful use within specific vulnerable social subgroups. As described above a systematic review found that, during economic crises, increased psychological distress and tighter budget constraints due to income reductions are the main behavioural mechanisms behind alcohol consumption and alcohol-related health problems. The former was found to lead to higher and more frequent alcohol consumption, and to a higher prevalence of drinking problems to cope with the distress; the latter can lead instead to lower alcohol consumption [19].

Although in Europe alcohol is the main contributor to chronic liver disease and cirrhosis mortality, other causes include hepatitis B or C infection and non-alcohol fatty liver disease, associated to obesity and type 2 diabetes. Pimpim reported that in this century viral hepatitis mortality increased in most Northern European countries, Hungary, Italy, Croatia and Portugal, but remained stable or decreased in others [26]. Ireland et al. observed a stable trend in mortality among individuals diagnosed with hepatitis C in England [31].

The fact that in our study chronic liver diseases and cirrhosis mortality was observed to be higher among men than women might be due to the higher alcohol consumption among men. Of note, a study of Bosque-Prous et al. [32] found that European countries with higher gender equality presented lower gender differences in hazardous drinking, mainly due to higher consumption among women. Due to this, the authors recommended policies aimed at improving gender equality to be accompanied by specific alcohol control policies.

We found socioeconomic inequalities in chronic liver diseases and cirrhosis mortality in men in the majority of cities, and in fourcities among women. These inequalities are related to the different factors associated with chronic liver diseases and cirrhosis, namely alcohol consumption, and hepatitis B and C. It is important to take into account that deprived areas have a higher proportion of vulnerable populations, such as drug users and marginalised groups, who can be at a greater risk of contracting hepatitis B and C [20], and exhibit higher alcohol consumption [33]. However, some reviews that have analysed area-level factors on alcohol use [33,34,35] did not find clear evidence that area-level disadvantage is associated with increased alcohol use. Previous studies have shown also socioeconomic inequalities in cirrhosis mortality in urban areas of European cities [15,36]. Moreover, studies that have analysed socioeconomic inequalities in alcohol-attributable mortality have found it is higher in lower socioeconomic groups in the majority of countries, and is similar in men and women [14,33,37,38]. Ford et al. described the increase of hepatitis C mortality in New York, from 2006 to 2014, and the highest rates occurred in neighbourhoods comprising socioeconomically disadvantaged groups [39].

Our study found that, during the 2008 economic crises, changes in the trends of socioeconomic inequalities in chronic liver diseases and cirrhosis mortality did not have statistical significance. Nevertheless, an increase was observed among men in Barcelona and Turin in the third period. Other studies analysing the trends in inequalities in cirrhosis mortality before the economic recession found different results: no increase for Barcelona [9] and an increase in Australia [12] The authors suggest that the increase in socioeconomic inequalities in Australia was due to the increased harmful alcohol consumption among lower socioeconomic groups, which might be attributed to a relative increased affordability of alcohol over time. Crombie and Precious described a reversal in the social class gradient of cirrhosis mortality during the 20th century (from advantaged classes to disadvantaged ones). These findings indicate a major change in risk factor distributions across social classes, possibly explained by differential changes in alcohol consumption [40]. The study by Mackenback et al. [14] found a rise in inequality in alcohol-related mortality through the years (1980–2009) in several countries in Eastern Europe (Hungary, Lithuania, and Estonia) and Northern Europe (Finland and Denmark). These findings can be explained by a rapid rise in alcohol-related mortality in lower socioeconomic groups. These trends should be monitored in the near future to observe their progress.

### Strengths and Limitations

This study reported, for the first time, chronic liver diseases and cirrhosis mortality and socioeconomic inequalities in 9 cities/metropolitan areas of Europe before and during the financial crisis. It is worth mentioning that vital statistics are a good source of information to study cirrhosis mortality in Europe [8] This study has some limitations. First, data used for socioeconomic indicators were dated from 2001, and the deprivation of areas could have changed over time; however, the rank of areas by socioeconomic deprivation has probably remained unchanged. Second, the number of areas included per city varied considerably, which may have affected the robustness of estimates (small number of deaths); we addressed this by using Bayesian methods to smooth the SMR of small areas. Third, the size of the small areas differed between the urban areas studied: smaller areas are more homogeneous and the possibility of observing stronger effects is higher, which may partly explain some of the higher RRs for Barcelona, Stockholm and Turin mainly among men.

## 5. Conclusions

This study shows a decrease in chronic liver diseases and cirrhosis mortality in men and women in European urban areas (except among men in Athens); the existence of geographical socioeconomic inequalities in this mortality; and the lack of significant change in these inequalities over time, after the onset of the financial crisis of 2008. However, these results should be monitored in the long term in order to study future changes.

Our findings hint at implementing policies to decrease risk factors for chronic liver diseases and cirrhosis, mainly among vulnerable populations. For alcohol consumption, we suggest policies such as: tax increase, the establishment of a minimum price for alcohol [24], marketing restrictions, and screening and behavioural interventions. Additionally, vaccination for hepatitis B for all population groups and the implementation of harm reduction programs for drug users should be prioritized [26].

## Figures and Tables

**Figure 1 ijerph-18-08801-f001:**
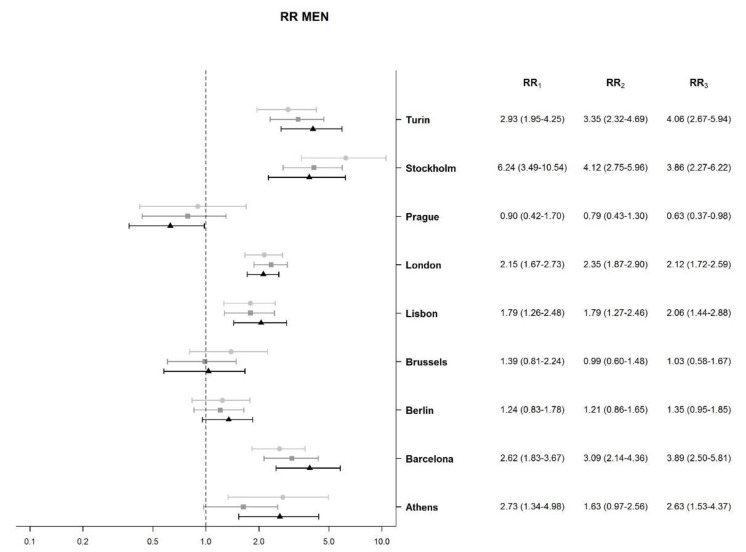
Association between socioeconomic deprivation index and liver cirrhosis mortality, relative risk (RR), and 95% credible intervals (CI) for men in nine urban areas. Notes: RR compares liver cirrhosis mortality of the 95th percentile value of socioeconomic deprivation (severe deprivation) to the 5th percentile value (low deprivation). RR_1_: RR in the first period (2000–2003); RR_2_: RR in the second period (2004–2008); RR_3_: RR in the third period (2009–2014).

**Figure 2 ijerph-18-08801-f002:**
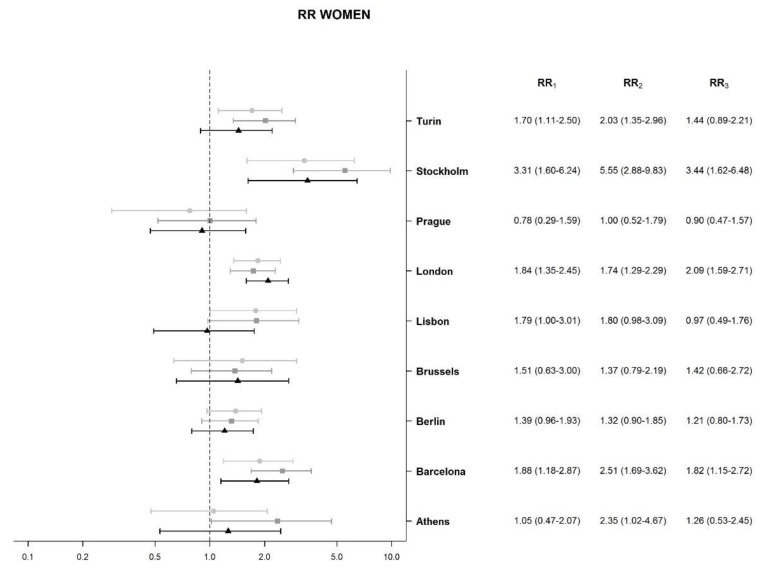
Association between socioeconomic deprivation index and liver cirrhosis mortality, relative risk (RR), and 95% credible intervals (CI) for women in nine urban areas. Notes: RR compares liver cirrhosis mortality of the 95th percentile value of socioeconomic deprivation (severe deprivation) to the 5th percentile value (low deprivation). RR_1_: RR in the first period (2000–2003); RR_2_: RR in the second period (2004–2008); RR_3_: RR in the third period (2009–2014).

**Table 1 ijerph-18-08801-t001:** Description of the nine European urban areas: number and type of small areas, period years, and total population (first- p25-, second -p50-, and third -p75- quartiles of the population by small area for men and women).

Urban Area	Short Name	Number of Small Areas	Type of Small Areas	Period Years			Population (First Year Available)							
				First Period 2000–2003	Second Period 2004–2008	Third Period 2009–2014	Men				Women			
							Total	p25	p50	p75	Total	p25	p50	p75
Athens metropolitan area	Athens	40	municipalities	2000–2003	2004–2008	2009–2013	1,577,172	18,565	29,745	35,489	1,710,446	20,136	32,163	39,965
Barcelona city	Barcelona	1491	census tracts	2000–2003	2004–2008	2009–2013	697,563	365	457	577	796,497	418	517	648
Berlin-Brandenburg metropolitan region	Berlin	30	districts	2002	2006	2011	2,927,616	66,326	96,176	129,157	3,047,188	68,041	97,454	130,560
Brussels capital region	Brussels	145	neighbourhoods	2001–2003	2004–2008	2009–2011	464,364	2727	4004	5707	505,673	3058	4288	6172
Lisbon metropolitan area	Lisbon	188	parishes	2000–2003	2004–2008	2009–2012	1275,813	2694	5437	8962	1,386,314	2938	5835	9904
Greater London	London	983	census tracts	2000–2003	2004–2008	2009–2014	3,597,120	3442	3810	4284	3,725,283	3526	3960	4382
Prague city	Prague	57	districts	2001–2003	2004–2008	2009–2014	549,652	1010	2206	15,001	610,466	1024	2100	14,838
Stockholm metropolitan area	Stockholm	1299	census tracts	2001–2003	2004–2008	2009–2011	897,487	218	560	1050	936,977	232	599	1104
Turin city	Turin	2678	census tracts	2000–2003	2004–2008	2009–2013	425,782	88	129	196	465,987	96	142	215

**Table 2 ijerph-18-08801-t002:** Number of deaths, crude mortality rate (MR), and indirectly standardised mortality rate (ISMR) per 100,000 inhabitants for men and women for each study period.

**Men**	**First Period (2000–2003)**			**Second Period (2004–2008)**			**Third Period (2009–2014)**		
**Urban Areas**	**Deaths**	**MR**	**ISMR**	**Deaths**	**MR**	**ISMR**	**Deaths**	**MR**	**ISMR**
Athens	489	7.8	8.83	595	7.85	8.5	671	9.08	9.42
Barcelona	564	19.58	19.66	604	15.91	16.36	566	14.71	14.88
Berlin	999	34.12	35.48	913	30.19	30.06	771	26.31	24.55
Brussels	189	-	-	386	-	-	207	-	-
Lisbon	1178	23.07	25.35	1154	17.87	19	862	16.25	16.73
London	1953	13.5	18.63	2369	12.65	17.41	2603	10.65	14.45
Prague	388	23.42	24.31	686	23.81	24.8	753	20.68	21.53
Stockholm	291	10.73	12.41	489	10.38	11.8	273	9.41	10.63
Turin	377	22.07	20.36	496	23.22	21.09	417	19.32	17.29
**Women**	**First Period**			**Second Period**			**Third Period**		
**Urban Areas**	**Deaths**	**MR**	**ISMR**	**Deaths**	**MR**	**ISMR**	**Deaths**	**MR**	**ISMR**
Athens	213	3.13	3.18	299	3.62	3.52	285	3.52	3.27
Barcelona	364	11.18	9.82	398	9.45	8.46	362	8.51	7.61
Berlin	497	16.31	15.25	467	14.99	13.69	422	13.82	12.07
Brussels	101	-	-	221	-	-	121	-	-
Lisbon	326	5.86	5.77	276	3.88	3.72	199	3.37	3.15
London	968	6.48	8.14	1069	5.53	7.06	1254	5.01	6.38
Prague	215	11.74	10.66	386	12.43	11.29	391	10.14	9.34
Stockholm	136	4.81	5.17	230	4.72	5.04	127	4.27	4.55
Turin	331	17.71	14.6	360	15.36	12.59	258	10.86	8.85

Note: Crude MR and ISMR are not displayed for Brussels, as population data were interpolated.

## Data Availability

Data can be shared upon request to the author.

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
