# Peer review of "Socioeconomic Inequalities in Chronic Liver Diseases and Cirrhosis Mortality in European Urban Areas before and after the Onset of the 2008 Economic Recession"

_ijerph, 2021, doi:10.3390/ijerph18168801_

Round 1

Reviewer 1 Report

This work addresses an interesting issue: to analyses the trends in chronic liver diseases and cirrhosis mortality, and in socioeconomic inequalities in this mortality, in nine European urban areas before and after the onset of the 2008 financial crisis.

The objectives are correctly defined. The methodology is well-designed and is consistent with the objectives of the study. The interpretation and discussion of results is clear, objective, and consistent, although a deeper analysis of the obtained data could be done. The conclusions summarize well the results obtained and are consistent with the work presented. However, there are some details that can be improved:

There is a lack of a more consistent foundation of the subject and supported by a more complete and in-depth review of the existing literature on the subject under analysis.

In section 2.3 Data Analysis, all the notation used in the expressions presented must be previously defined. Expressions must be properly numbered.

You may want to review the formatting of the last paragraph of the Conclusions (lines 298-302).

Author Response

This work addresses an interesting issue: to analyses the trends in chronic liver diseases and cirrhosis mortality, and in socioeconomic inequalities in this mortality, in nine European urban areas before and after the onset of the 2008 financial crisis.

The objectives are correctly defined. The methodology is well-designed and is consistent with the objectives of the study. The interpretation and discussion of results is clear, objective, and consistent, although a deeper analysis of the obtained data could be done. The conclusions summarize well the results obtained and are consistent with the work presented. However, there are some details that can be improved:

There is a lack of a more consistent foundation of the subject and supported by a more complete and in-depth review of the existing literature on the subject under analysis.

Answer: We have improved the review and added new bibliographic references

In section 2.3 Data Analysis, all the notation used in the expressions presented must be previously defined. Expressions must be properly numbered.

Answer: Thank you very much for your comment. We now define all the notations presented, have improved equations with the word editor, and numbered the expressions accordingly.

You may want to review the formatting of the last paragraph of the Conclusions (lines 298-302).

Answer: We have reviewed it

Reviewer 2 Report

The paper can be summarized as follows. The goal of this research was to look at trends in chronic liver disease and cirrhosis mortality, as well as socioeconomic disparities in this mortality, in nine European cities before and after the financial crisis of 2008. Methods: This is an ecological study of patterns in three time periods before and after the economic crisis (2000-2008). (2009-2014). The little areas of nine European cities served as the study's units. We looked at cirrhosis Standardized Mortality Ratios by city, region, and gender, which were smoothed using a hierarchical Bayesian model.  The socioeconomic deprivation index, the period, and their interaction were all incorporated in an ecological regression model that was used to analyze the trend in socioeconomic inequality.
During the financial crisis, mortality rates of liver cirrhosis declined in all cities, with the exception of men in Athens (rates increased from 8.50 per 100,000 inhabitants during the second period to 9.42 during the third). Men have a higher mortality risk from cirrhosis than women. Cirrhosis mortality disparities were discovered in 6 cities/metropolitan areas among males and 4 cities/metropolitan areas among women. Despite the fact that changes in inequality among men were not statistically significant, they did increase in Turin and Barcelona. Several cities exhibited reduced inequality among women in the third period. During the 2008 financial crisis, there were no changes in spatial socioeconomic inequalities in liver cirrhosis, which primarily affect men. These results, however, should be closely examined in the long run.

My evaluation of the paper is as follows.  The subject is of great interest and I judge the study as of great quality. I formulate the following minor remarks that dont deserve another round (for me): 

o Please improve the English. It misses punctuations, some grammar mistakes are here, etc. The paper deserves a better English.

o CI is for credible interval or confidence interval ?

o Improve a bit the presentation of the Equations of the models.

Author Response

The paper can be summarized as follows. The goal of this research was to look at trends in chronic liver disease and cirrhosis mortality, as well as socioeconomic disparities in this mortality, in nine European cities before and after the financial crisis of 2008. Methods: This is an ecological study of patterns in three time periods before and after the economic crisis (2000-2008). (2009-2014). The little areas of nine European cities served as the study's units. We looked at cirrhosis Standardized Mortality Ratios by city, region, and gender, which were smoothed using a hierarchical Bayesian model.  The socioeconomic deprivation index, the period, and their interaction were all incorporated in an ecological regression model that was used to analyze the trend in socioeconomic inequality.
During the financial crisis, mortality rates of liver cirrhosis declined in all cities, with the exception of men in Athens (rates increased from 8.50 per 100,000 inhabitants during the second period to 9.42 during the third). Men have a higher mortality risk from cirrhosis than women. Cirrhosis mortality disparities were discovered in 6 cities/metropolitan areas among males and 4 cities/metropolitan areas among women. Despite the fact that changes in inequality among men were not statistically significant, they did increase in Turin and Barcelona. Several cities exhibited reduced inequality among women in the third period. During the 2008 financial crisis, there were no changes in spatial socioeconomic inequalities in liver cirrhosis, which primarily affect men. These results, however, should be closely examined in the long run.

My evaluation of the paper is as follows.  The subject is of great interest and I judge the study as of great quality. I formulate the following minor remarks that dont deserve another round (for me): 

o Please improve the English. It misses punctuations, some grammar mistakes are here, etc. The paper deserves a better English.

Answer: the article has been reviewed again by a person specialized in correcting and editing manuscripts written in English.

o CI is for credible interval or confidence interval ?

Answer: Thank you very much for your comment. CI is for credible interval as mentioned in line 55 and in the title of the figures.

o Improve a bit the presentation of the Equations of the models.

Answer: Thank you very much for your comment. We have improved the presentation of the equations by using the word equations editor.